# Identification of Efflux Pump Mutations in *Pseudomonas aeruginosa* from Clinical Samples

**DOI:** 10.3390/antibiotics12030486

**Published:** 2023-03-01

**Authors:** Sonia Quddus, Zainab Liaqat, Sadiq Azam, Mahboob Ul Haq, Sajjad Ahmad, Metab Alharbi, Ibrar Khan

**Affiliations:** 1Centre of Biotechnology and Microbiology, University of Peshawar, Peshawar 25120, Pakistan; 2Department of Pharmacy, Abasyn University, Peshawar 25000, Pakistan; 3Department of Computer Science and Physics, Center for Soft Matter and Biological Physics, Virginia Tech, Blacksburg, VA 24060, USA; 4Department of Health and Biological Sciences, Abasyn University, Peshawar 25000, Pakistan; 5Department of Pharmacology and Toxicology, College of Pharmacy, King Saud University, P.O. Box 2455, Riyadh 11451, Saudi Arabia

**Keywords:** *Pseudomonas aeruginosa*, antibiotic-resistant efflux pump genes, nosocomial pathogen, antibiotics susceptibility pattern

## Abstract

Efflux pumps are a specialized tool of antibiotic resistance used by *Pseudomonas aeruginosa* to expel antibiotics. The current study was therefore conducted to examine the expression of MexAB-OprM and MexCD-OprJ efflux pump genes. In this study, 200 samples were collected from Khyber Teaching Hospital (KTH) and Hayatabad Medical Complex (HMC) in Peshawar, Pakistan. All the isolates were biochemically identified by an Analytical Profile Index kit and at the molecular level by Polymerase Chain Reaction (PCR) utilizing specific primers for the OprL gene. A total of 26 antibiotics were tested in the current study using the guidelines of the Clinical and Laboratory Standard Institute (CLSI) and high-level resistance was shown to amoxicillin-clavulanic acid (89%) and low-level to chloramphenicol (1%) by the isolates. The antibiotic-resistant efflux pump genes MexA, MexB, OprM, MexR, MexC, MexD, OprJ, and NfxB were detected in 178 amoxicillin-clavulanic acid-resistant isolates. Mutations were detected in MexA, MexB, and OprM genes but no mutation was found in the MexR gene as analyzed by I-Mutant software. Statistical analysis determined the association of antibiotics susceptibility patterns by ANOVA: Single Factor *p* = 0.05. The in silico mutation impact on the protein structure stability was determined via the Dynamut server, which revealed the mutations might increase the structural stability of the mutants. The docking analysis reported that MexA wild protein showed a binding energy value of −6.1 kcal/mol with meropenem and the mexA mutant (E178K) value is −6.5 kcal/mol. The mexB wild and mutant binding energy value was −5.7 kcal/mol and −8.0 kcal/mol, respectively. Efflux pumps provide resistance against a wide range of antibiotics. Determining the molecular mechanisms of resistance in *P. aeruginosa* regularly will contribute to the efforts against the spread of antibiotic resistance globally.

## 1. Introduction

*Pseudomonas aeruginosa* is a predominant Gram-negative, aerobic, motile rod belonging to the family Pseudomonadaceae [1]. *P. aeruginosa* is present in soil and water and is a well-known pathogen causing diseases in humans, animals, and plants. Due to pigment production, pyoverdine, pyocyanin, and pyorubin by *P. aeruginosa* are easily detected on agar plates [2]. In comparison to other bacteria, the genome size of *P. aeruginosa* is very large (5.5–7 Mbp) and encodes many regulatory proteins/enzymes important for metabolism, development, and efflux system (hence for antibiotic resistance). Due to this huge encoding ability, *P. aeruginosa* becomes more stable and adapts to a variety of harsh environments [3]. *P. aeruginosa* is ubiquitous and causes severe infections in immunocompromised individuals. It causes healthcare-associated infections including sepsis, respiratory tract infections, hospital-acquired pneumonia, urinary tract infections, skin infections, bacterial keratitis, bacterial colitis, and otitis externa [4]. The treatment for the infections caused by *P. aeruginosa* includes mono and combination therapy [5]. The combination therapy may reduce the mortality rate in patients infected with *P. aeruginosa*. However, the well-documented antibiotic-resistant mechanisms of *P. aeruginosa* to a wide range of antibiotics are the main hurdle in treatment. Moreover, the over and misuse of antibiotics are responsible for antibiotic resistance in *P. aeruginosa* which is often multidrug resistant. *P. aeruginosa* has developed resistance against major antibiotic families including β-lactams, aminoglycosides, quinolones, and carbapenem [6]. The resistance mechanisms include adaptive resistance, acquired resistance, and intrinsic resistance [7]. The formation of biofilm protects against many antibiotics and contributes to the adaptive resistance of *P. aeruginosa* [8]. The antibiotic resistance genes can be acquired from the environment by *P. aeruginosa* via horizontal gene transfer and mutations are further adding to the phenomenon of acquired resistance [9]. The overexpression of efflux pumps diminished outer membrane permeability, and the production of enzymes for inactivating antibiotics are the main contributors to the intrinsic resistance of *P. aeruginosa* [10]. The efflux pumps of the Resistant Nodulation Division (RND) family are among the main efflux pumps of *P. aeruginosa* which contribute to resistance to many antibiotics. The MexAB-OprM is the first efflux pump detected in *P. aeruginosa*, regulated by the *mexR* gene, and is able to expel a wide range of antibiotics such as β-lactams, fluoroquinolones, tetracycline, macrolides, β-lactamase inhibitors, chloramphenicol, and sulfonamides. The efflux pump MexCD-OprJ, regulated by the nfxB gene is similar to the MexAB-OprM efflux pump [11]. Other efflux pumps such as MexEF-OprN and MexXY-OprM show resistance to a narrower spectrum of antibiotics [12]. There is a need to investigate the role of efflux pumps in clinical isolates of *P. aeruginosa* so that appropriate strategies and antibiotics can be used to manage the respective diseases. The current study focused on the expression and mutations of MexAB-OprM and MexCD-OprJ efflux pumps in clinical isolates of *P. aeruginosa* and correlated the expression of genes with antibiotic susceptibility profiles of *P. aeruginosa*.

## 2. Materials and Methods

### 2.1. Isolation and Identification of Bacterial Isolates

The current research was carried out at the Molecular Microbiology laboratory of the Centre of Biotechnology and Microbiology (COBAM), University of Peshawar.

A total of 200 clinical samples of *P. aeruginosa* were collected, of which 52 were from the Pathology and Microbiology laboratory of Khyber Teaching Hospital (KTH) Peshawar and 148 from the Hayatabad Medical Complex (HMC) Peshawar. All the samples were inoculated on nutrient agar and MacConkey agar plates and were incubated at 37 °C for 24 h for bacterial growth. After incubation, bacterial colonies were subjected to phenotypic and genotypic identification. The phenotypic identification was carried out by Gram staining to determine the Gram-negative status of the bacteria [13]. For biochemical identification, Analytical Profile Index (API 20E) strips were used [14].

### 2.2. Extraction of Genomic DNA

After the identification of isolates, 24 h old bacterial cultures were used for the extraction of bacterial DNA via a GJC^®^DNA purification kit. After DNA extraction, DNA samples were run on 1.5% agarose gel and visualized under Bio-Rad Molecular Imager^®^ Gel Doc™.

### 2.3. Molecular Identification of Bacterial Isolates

For confirmation of isolates, genotypic identification was performed via the *oprL* gene by using a specific primer under optimized PCR conditions (Table 1) After PCR, the PCR product was run on 1.5% agarose gel and visualized under Bio-Rad Molecular Imager^®^ Gel Doc™.

### 2.4. Antibiotic Susceptibility Testing

The antibiotic susceptibility pattern of the identified isolates was performed by the Kirby Bauer disc diffusion method against selected antibiotics (Table 2) as prescribed by the Clinical and Laboratory Standards Institute (CLSI) 2019. Sterile plates of Muller Hinton Agar (MHA) were prepared, and selected antibiotic discs were placed and incubated for 24 h at 37 °C. The zones of inhibition were measured and interpreted as susceptible, intermediate, and resistant according to the CLSI guidelines [15].

### 2.5. Molecular Detection of Efflux Pump Resistance Genes

The efflux pump-resistant genes *MexA-MexB-OprM* and *MexC-MexD-OprJ*, with regulators *mexR* and *nfxB*, respectively, were investigated in all isolates by PCR. The PCR mixture was prepared by adding 12.5 µL GoTaq^®^ Green Master Mix 2X, 1 µL upstream primer, 1 µL downstream primer, 25 µL PCR grade water, and 1 µL DNA template and run under optimized conditions (Table 1). After that, samples were run on 1.5% agarose gel and visualized under the gel documentation system.

### 2.6. Mutational Analysis of PCR Products

After the amplification of efflux pump-resistant genes, PCR products were sent to Macrogen for sequencing using the next-generation sequencing (NGS) method. The sequences of genes were analyzed through the BioEdit Sequence Alignment Editor Software (Borland, Vista, CA, USA). The consensus sequence of each gene was checked through the Basic Local Alignment Search Tool (BLAST) which checked the local similarity between the sequences. Interpretation of I-mutant results was used to predict either an increase or decrease in the function of the respective proteins.

### 2.7. Computational Studies

By using the Expasy translater tool (https://web.expasy.org/translate/ accessed on 8 September 2022), the nucleotide sequences of the genes were converted into amino acid sequences to be used for structure modeling and docking studies. The SWISS-MODEL server (https://swissmodel.expasy.org/) was used for the structural modeling of wild and mutant proteins. SWISS-MODEL accepts the protein sequence in FASTA format. The protein structure visualization was performed through UCSF Chimera v1.16 (http://www.cgl.ucsf.edu/chimera/ accessed on 15 September 2022). The mutation effect on the protein structure and overall conformational stability was determined by the Dynamut server available at https://biosig.lab.uq.edu.au/dynamut/prediction accessed on 20 September 2022. The PyRx 0.8 virtual screening software (https://pyrx.sourceforge.io/ accessed on 25 September 2022) was used for molecular docking studies to determine the intermolecular binding conformation of wild and mutant proteins with meropenem. The docking was performed on Intel^®^ Core(TM) i5-3230M CPU @ 2.60 GHz with 64-bit Windows 8.1. The grid box dimensions were set manually to cover the whole protein. For mexA wild-type protein, the dimensions were x = 346.21 Å, y = 317.80 Å, and z = 333.04 Å. The docking dimensions for the mexA mutant were set to 74.19 Å on x = 342.03 Å, 282.35 Å on the *y*-axis, and 329.09 Å on the *z*-axis. The box dimensions for mexB wild were set to 79.64 Å on the *x*-axis, −45.72 Å on the *y*-axis, and −17.71 Å on the z-axis. For the mexB mutant, the dimensions used were x = −34.72 Å, *y*-axis = −22.56 Å, and *z*-axis = 20.64 Å. The docking complexes were analyzed by UCSF Chimera v1.16 and Discovery Studio (DS) Visualizer v2021.

### 2.8. Statistical Analysis

A chi-square analysis was conducted using SPSS version 20 to find the association between the expected value of *E. coli* with the observed *p* ≤ 0.05. For that, the number of samples was (*n*) set at 150 and the degree of freedom was taken at n-1. For comparative analysis, one-way analysis of variance (ANOVA) among the continuous values of antibiotics with *P. aeruginosa* was performed and *p* ≤ 0.05 values were considered statistically significant.

## 3. Results

The clinical isolates of *P. aeruginosa* were collected from the KTH and the HMC, Peshawar, from different sources: wound swab, urine, pus, blood, ear pus, and cerebrospinal fluid (Table 3). One hundred and eight patients (54%) were male and 92 (46%) were female and of different age groups. Among 200 isolates of *P. aeruginosa*, a high rate of prevalence was recorded in the age group of 21 to 30 (21.5%) followed by the age group of 31 to 40 (18.5%) (Table 4).

### 3.1. Antibiotics Susceptibility Testing

The antibiotic sensitivity pattern of the isolates revealed sensitivity to AK, SCF, and TZP and high resistance to AMC, CTX, CFM, and SXT (Table 5)

### 3.2. Molecular Detection of Efflux Pump Resistance Genes in Isolates of P. aeruginosa

The PCR results revealed the presence of efflux pump genes in *P. aeruginosa* isolates (Figure 1, Figure 2, Figure 3, Figure 4, Figure 5, Figure 6, Figure 7 and Figure 8). By comparing the results of PCR with the antibiotic susceptibility pattern of isolates, it was concluded that efflux pump resistance genes were detected mostly among amoxicillin/clavulanic acid-resistant isolates (*n* = 178; 89%) (Table 6).

### 3.3. Mutational Analysis of Antibiotic-Resistant Efflux Pump Genes

The mutational analysis was performed for the *mexA*, *mexB*, *oprM,* and *mexR* genes. In the sequences of *mexA (*Table 7 and Table 8), *mexB* (Table 9 and Table 10), and *oprM* gene (Table 11 and Table 12) mutations were detected while no mutation was detected in the *mexR* gene.

### 3.4. Mutation Impact on Structure Stability

The impact of mutations on the thermodynamic characteristics of wild-type and mutant proteins was revealed through the Dynamut server. The Dynamut predicts each mutation’s impact on protein conformational energy. As given in Appendix A, the mutation effect determines the increased stability of mutant proteins compared to wild proteins. The E178K of mexA showed a destabilizing effect. In case of mexB, mutations such as R2T, W4T, L5V, D6T, P7F, A8E, N9Q, L10G, N11T, S12D, Y13P, Q14D, L15I, T16A, P17Q, G18V, D19Q, S21Q, S22N, A23K, I24L, H25Q, A26L, Q27A, N28T, V29P, Q30L, I31L, S32P, S33Q, G34E, Q35V, L36Q, G37R, G38Q, L39G, P40I, N43T, G44K, Q45A, H46V, L47K, A49F, T50L, I51M, I52V, G53V, K54G, T55V, R56V, L57S, Q58T, T59D, A60G, E61S, Q62M, F63T, E64K, N65E, I66D, L68S, K69N, V70Y, N71I, P72V, D73S, G74N, S75I, V77D, R78P, K80S, D81R, V82T, A83K, D84G, L87D, G88F, G89Q, H90V, D91F, Y92G, I94Q, N95Y, A96R, Q97S, F98M, N99R, G100I, S101W, P102L, G103D, V104P, R105A, Y106K, R107L, D108N, Q109S, and A110Y reported a destabilizing effect on the wild mexB protein. The vibrational entropy energy between the wild and mutant types was recorded in kcal/mol. 

### 3.5. Docking Analysis

Molecular docking is a computational-based technique for intermolecular binding conformation. Here, the objective was to determine the mutation impact on meropenem drug binding with wild and mutant phenotypes of the genes. The docking results are provided in Table 13. The mexA wild protein complex binding energy value was −6.1 kcal/mol and the mexA mutant (E178K) value was −6.5 kcal/mol. The mexB wild protein complex binding energy value was −5.7 kcal/mol and the mexB mutant protein complex binding energy value was −8.0 kcal/mol. The binding conformation of meropenem with the mexA and mexB is shown in Figure 9 and Figure 10.

Through discovery studio visualizer v2021 software, the binding interactions between protein and drug were determined. The wild-type MexA is involved in van der Waals and conventional hydrogen bonds with the drug, while the mutant formed van der Waals conventional hydrogen, and carbon-hydrogen bonds. The wild MexA active residue such as Arg35 is attached to the hydroxybutanal with the help of a conventional hydrogen bond while 1-azabicyclo[3.2.0]hept-2-ene of the drug produced chemical bonding with Ala40, Gly37, Ala 36, Gly99, Glu58, Lue96, Leu28, Leu24, Arg25, leu21, Phe61, Val64, and Ile75. In mutant MexA, Lys173 is attached to the 1-azabicyclo[3.2.0]hept-2-ene through a conventional hydrogen bond. The val175 is attached to the 1-azabicyclo[3.2.0]hept-2-ene with the help of a carbon-hydrogen bond. The active residues such as Pro176,Thr160, Ala177, Glu161, Phe165,Val 166,Ile158, Lys157, The174, Val125, Ile159, Val172, and Gly162 were engaged with 1-azabicyclo [3.2.0]hept-2-ene by Van der Waals bonding (Figure 11). In mexB wild, binding interactions involve Arg2 and Ile3 attached to the 1-azabicyclo[3.2.0]hept-2-ene-2-carboxylic acid by a conventional hydrogen bond. The Asn28 is attached to the pyrrolidine-2-carboxamide chemical moiety via a conventional hydrogen bond. The active residues such as Pro17, Val20, Phe63, Ser21, Leu5, His25, Ile24, Arg56, Trp4, and Met1 interact with the drug by van der Waals interactions (Figure 12). The mexB mutant binding interactions involve Thr11, Gly10, and Phe7 with the pyrrolidine-2-carboxamide through conventional hydrogen bonding while Val5 is seen with 1-azabicyclo[3.2.0]hept-2-en-7-one. The active site residues Val42, Gln17, Ala16, Pro13, Glu8, Gln9, Thr208, Thr6, Ala45, Thr4, Lys44, Lys44, Thr43, and Val20 formed bonding to the protein via van der Waals interactions.

## 4. Discussion

A recent study investigated the expression of the *MexA* (88.2%) and *MexB* genes (70.5%) in 136 MDR and PDR isolates of *P. aeruginosa*. The study reported 69% *MexB* gene expression followed by 28.7% *MexC* expression, 43.4% *MexE* expression, and 74.6% *MexY* expression among isolates from the ICU. They were highly resistant to ticarcillin (80%), ciprofloxacin (74%), and meropenem (71%) [13].

In another study, antibiotic resistance-conferring efflux pumps were investigated in the isolates that were carbapenem-resistant (63.15%). The PCR results revealed overexpression in 19 (79.1%) isolates [14]. In the present study, MexAB-OprM and MexCD-OprJ efflux pumps were expressed in all the amoxicillin/clavulanic acid-resistant isolates. Mohseni et al., [15] investigated the efflux pumps conferring resistance among isolates collected from both human and animal sources. The PCR results showed an increased expression of the MexA gene as compared to the MexB gene. The isolates were 100% resistant to trimethoprim/sulfamethoxazole, cefazolin, ampicillin, kanamycin, and amoxicillin/clavulanic acid.

Efflux pump systems also mediate fluoroquinolone resistance in *P. aeruginosa*. In another study, out of 36 isolates, 88% were resistant to ofloxacin while 85% of them were resistant to sparfloxacin. Thus, the resistance mediated by efflux pump systems must be considered when introducing novel fluoroquinolones [16]. A study by Rudy et al. detected the expression of MexA-MexB-OprM efflux pump in 80% of isolates that were all ciprofloxacin resistant [17]. In the current study, 79 (39.5%) isolates were resistant to ciprofloxacin. The *MexA, MexB, OprM,* and *MexR* genes were detected in these ciprofloxacin-resistant isolates in accordance with the reported literature [18,19].

## 5. Conclusions

*P. aeruginosa* is known to adapt efficiently in harsh environments. All isolates in the present study were highly resistant to various families of antibiotics including beta-lactams, aminoglycosides, tetracycline, and carbapenems. Among 200 isolates, 178 were highly resistant and expressed all the selected efflux pump-resistant genes. For the better treatment of infections by *P. aeruginosa*, combination therapies may be a good choice to overcome the multidrug-resistant mechanisms of *P. aeruginosa*.

## 6. Future Recommendations

All isolates in the present study were highly resistant showing expression of efflux pumps. To overcome this hurdle, the implementation of efflux pump inhibitors with antibiotics would be helpful. Research for novel antibiotics and efflux pump inhibitors could be an interesting strategy for the better management of infections caused by *P. aeruginosa*.

## Figures and Tables

**Figure 1 antibiotics-12-00486-f001:**
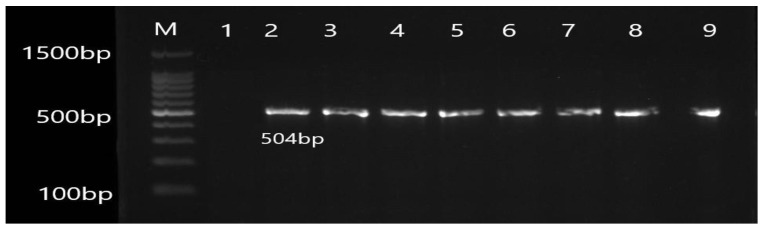
Electrophoresis showing amplicons of *P. aeruginosa mexB* gene. Lane M: 100 bp plus molecular marker, Lane 1: Negative control, Lane 2–9: Positive isolates of *mexB* gene.

**Figure 2 antibiotics-12-00486-f002:**
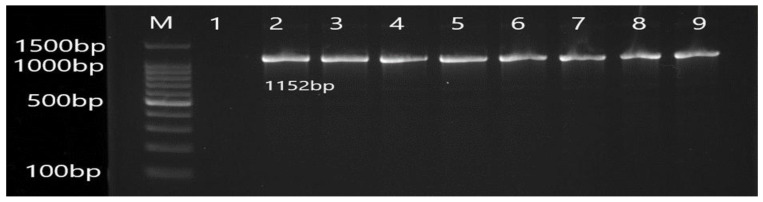
Electrophoresis showing amplicons of *P. aeruginosa mexA* gene. Lane M: 100 bp molecular marker, Lane 1: Negative control, Lane 2–9: Positive isolates of *mexA* gene.

**Figure 3 antibiotics-12-00486-f003:**
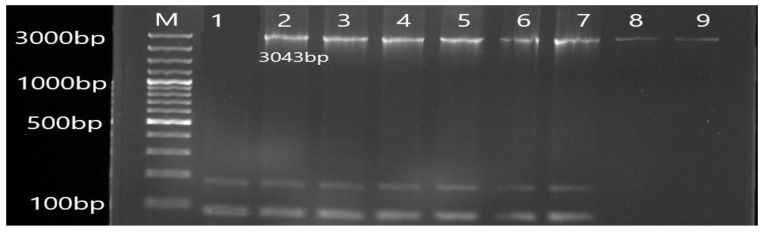
Electrophoresis showing amplicons of *P. aeruginosa oprL* gene. Lane M: 100 bp molecular marker, Lane 1: Negative control, Lane 2: Positive control, Lane 3–9: Positive isolates of *oprL*.

**Figure 4 antibiotics-12-00486-f004:**
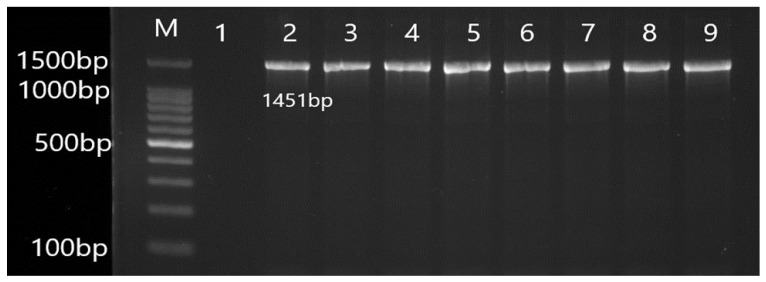
Electrophoresis showing amplicons of *P. aeruginosa mexC* gene. Lane M: 100 bp molecular marker, Lane 1: Negative control, Lane 2–9: Positive isolates of *mexC* gene.

**Figure 5 antibiotics-12-00486-f005:**
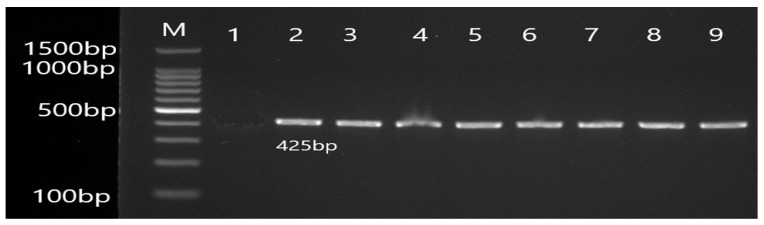
Electrophoresis showing amplicons of *P. aeruginosa mexR* gene. Lane M: 100 bp molecular marker, Lane 1: Negative control, Lane 2–9: Positive isolates of *mexR* gene.

**Figure 6 antibiotics-12-00486-f006:**
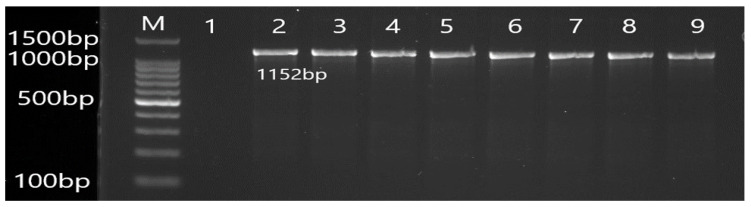
Electrophoresis showing amplicons of *P. aeruginosa oprM* gene. Lane M: 100 bp molecular marker, Lane 1: Negative control, Lane 2–9: Positive isolates of *oprM* gene.

**Figure 7 antibiotics-12-00486-f007:**
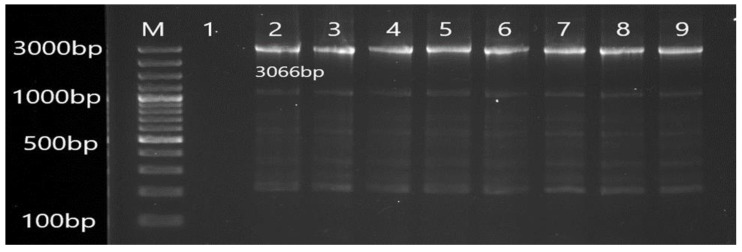
Electrophoresis showing amplicons of *P. aeruginosa mexD* gene. Lane M: 100 bp plus molecular marker, Lane 1: Negative control, Lane 2–9: Positive isolates of *mexD* gene.

**Figure 8 antibiotics-12-00486-f008:**
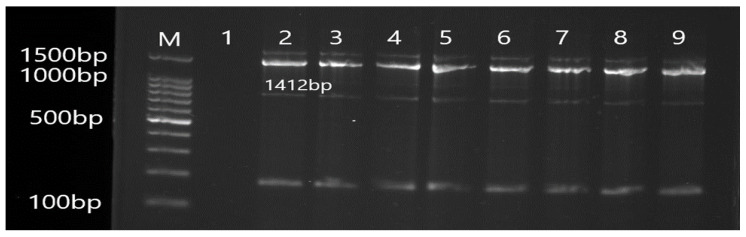
Electrophoresis showing amplicons of *P. aeruginosa oprJ* gene. Lane M: 100 bp molecular marker, Lane 1: Negative control, Lane 2–9: Positive isolates of *oprJ* gene.

**Figure 9 antibiotics-12-00486-f009:**
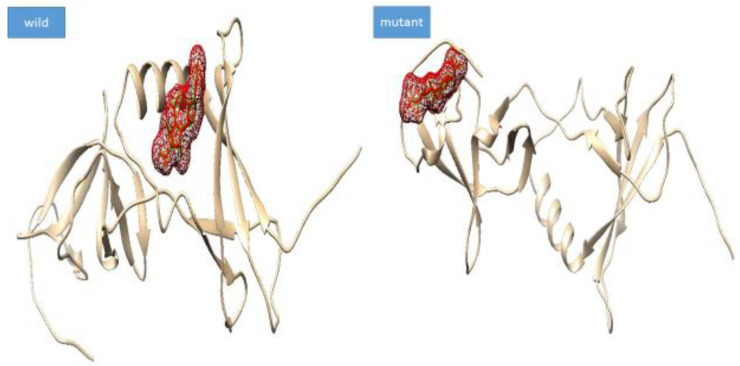
MexA wild and mutant intermolecular-docked complex with meropenem. The proteins are shown in tan cartoon style while the ligands are given in mesh.

**Figure 10 antibiotics-12-00486-f010:**
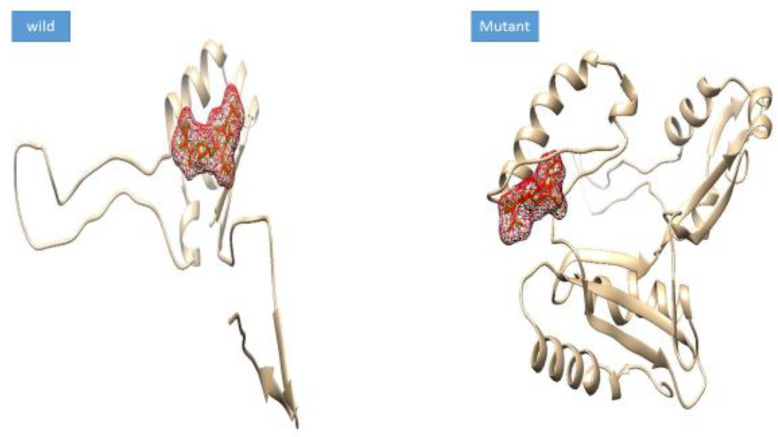
Binding conformation of meropenem with the mexB wild and mutant proteins. The proteins are shown in tan cartoon style while the ligands are given in mesh.

**Figure 11 antibiotics-12-00486-f011:**
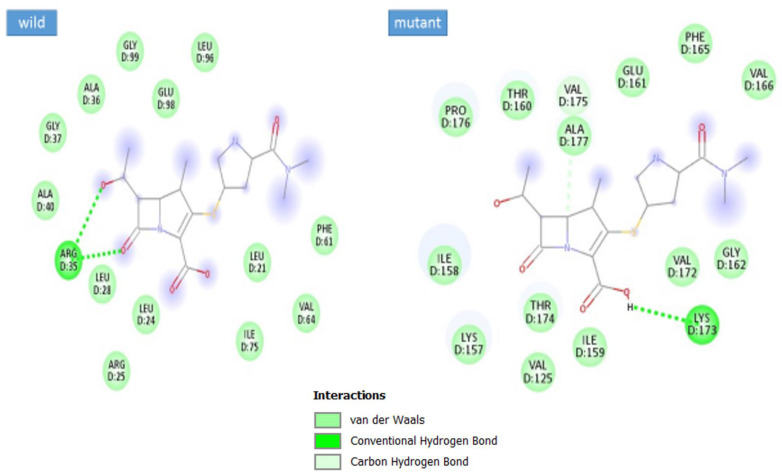
MexA wild and mutant binding interactions with meropenem. The compound is presented in a 2D line.

**Figure 12 antibiotics-12-00486-f012:**
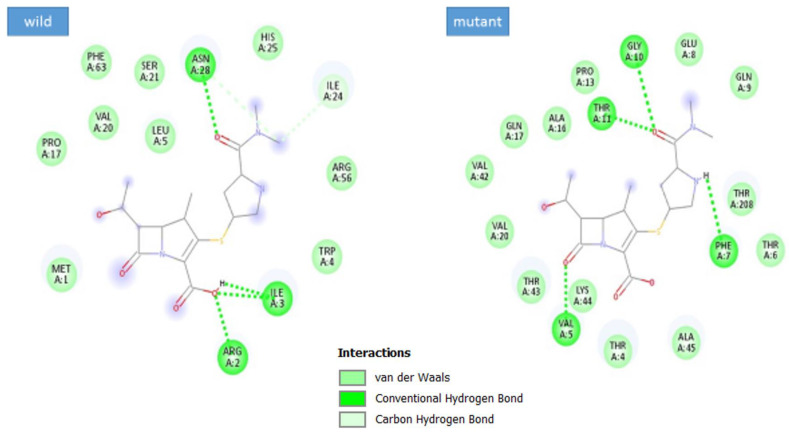
MexB wild and mutant binding interactions with meropenem. The compound is presented in a 2D line.

**Table 1 antibiotics-12-00486-t001:** Primer sequences with optimized PCR conditions.

Gene	Primer	Product Size (bp)	Annealing Temperature (°C)
** *OprL* **	**F** ATGGAAATGCTGAAATTCGGC **R** CTTCTTCAGCTCGACGCGACG	504	55
** *MexA* **	**F** CTATGCAACGAACGCCAGC **R** AGCCCTTGCTGTCGGTTTTC	1152	56
** *MexB* **	**F** TAGGCCCATTTTCGCGTGG **R** CGGTACCCAGAAGATCGCC	3043	56
** *OprM* **	**F** CGGTCCTTCCTTTCCCTGG **R** CAAGCCTGGGGATCTTCCTT	1451	55
** *MexR* **	**F** CAAGCGGTTGCGCGG **R** CCCCGTGAATCCCGACCTG	425	56
** *MexC* **	**F** TTACTGTTGCGGCGCAGG **R** CGTGCAATAGGAAGGATCGG	1152	55
** *MexD* **	**F** CAGCAGCCAGACGAAACAGA **R** TTCTTCATCAAGCGGCCGAA	3066	56
** *OprJ* **	**F** CTGCCGCCTCGATGTACC **R** GTATCGGCGCTGCTGATCG	1412	55
** *NfxB* **	**F** GACCCTGATTTCCCATGACG **R** GGAACATCTGCTCCAGGGTAT	530	56

**Table 2 antibiotics-12-00486-t002:** List of antibiotics.

S. No	Antibiotics (µg)	Family (Symbol)
**1**	Amikacin (20)	Aminoglycoside (AK)
**2**	Gentamicin (10)	Aminoglycoside (CN)
**3**	Azithromycin (30)	Macrolide (AZM)
**4**	Tigecycline (15)	Tetracycline (TGC)
**5**	Chloramphenicol (30)	Chloramphenicol (C)
**6**	Ciprofloxacin (5)	Fluoroquinolone (CIP)
**7**	Levofloxacin (5)	Fluoroquinolone (LEV)
**8**	Moxifloxacin (5)	Fluoroquinolone (MXF)
**9**	Amoxicillin (25)	β-lactam (penicillin) (AML)
**10**	Amoxicillin-clavulanic acid (30)	β-lactam (penicillin) (AMC)
**11**	Piperacillin-tazobactam (110)	β-lactam (penicillin) (TZP)
**12**	Aztreonam (30)	β-lactam (monobactams) (ATM)
**13**	Cefotaxime (30)	β-lactam (cephalosporin) (CTX)
**14**	Cefepime (30)	β-lactam (cephalosporin) (FEP)
**15**	Ceftazidime (30)	β-lactam (cephalosporin) (CAZ)
**16**	Cefoperazone (75)	β-lactam (cephalosporin) (CFP)
**17**	Cefoperazone-sulbactam (105)	β-lactam (cephalosporin) (SCF)
**18**	Ceftriaxone (30)	β-lactam (cephalosporin) (CRO)
**19**	Cefixime (5)	β-lactam (cephalosporin) (CFM)
**20**	Meropenem (10)	β-lactam (carbapenem) (MEM)
**21**	Imipenem (10)	β-lactam (carbapenem) (IMP)
**22**	Fosfomycin (50)	Fosfomycin (FOS)
**23**	Colistin (10)	Polymyxin (CT)
**24**	Polymyxin B (300)	Polymyxin (PB)
**25**	Trimethoprim-sulfamethoxazole (25)	Sulfonamide (SXT)
**26**	Nitrofurantoin (300)	Nitrofurantoin (F)

**Table 3 antibiotics-12-00486-t003:** Collection of clinical samples of *P. aeruginosa* from various sources.

Source	Number (Percentage)
Urine catheter	1 (0.5)
Stone analysis	1 (0.5)
Urine	28 (14)
Pus	57 (28.5)
Wound swab	94 (47)
Blood	7 (3.5)
Sputum	9 (4.5)
CSF	1 (0.5)
Ear swab	2 (1.0)
Total	200

**Table 4 antibiotics-12-00486-t004:** Frequency of patients’ gender and age.

Parameter		Frequency	Percentage
Gender	Male	108	54.0
	Female	92	46.0
Age Group (Years)	1–10	12	6
	11–20	30	15
	21–30	43	21.5
	31–40	37	18.5
	41–50	23	11.5
	51–60	25	12.5
	61–70	21	10.5
	71–80	8	4
	81–90	1	0.5

**Table 5 antibiotics-12-00486-t005:** Antibiotic susceptibility pattern of *P. aeruginosa*.

Antibiotics	Resistant (*n*)	Percentage (%)	Intermediate (*n*)	Percentage (%)	Susceptible (*n*)	Percentage (%)
AK	40	20	4	2	156	78
CN	88	44	10	5	102	51
CIP	79	39.5	9	4.5	112	58
LEV	71	35.5	23	11.5	106	53
MXF	80	40	11	5.5	109	54.5
AML	6	3	-	-	1	0.5
AMC	178	89	1	0.5	21	10.5
TZP	49	24.5	5	2.5	146	73
ATM	71	35.5	16	8.0	113	56.5
CTX	128	64	5	2.5	67	33.5
FEP	72	36	7	3.5	121	60.5
CAZ	73	36.5	11	5.5	116	58
CEP	72	36	15	7.5	113	56.5
SCF	49	24.5	10	5.0	141	70.5
CRO	96	48	11	5.5	93	46.5
CFM	158	79	7	3.5	35	17.5
MEM	63	31.5	8	4.0	129	64.5
IMP	63	31.5	11	5.5	126	63
AZM	-	-	-	-	7	3.5
TGC	100	50	12	6	88	44
CT	62	31	17	8.5	121	60.5
PB	63	31.5	21	10.5	116	58
FOS	6	3	2	1	22	11
C	2	1	-	-	5	2.5
SXT	125	62.5	5	2.5	70	35
F	15	7.5	-	-	15	7.5

**Table 6 antibiotics-12-00486-t006:** Polymerase chain reactions of Antibiotic resistance efflux pump genes.

Positive Isolates of Efflux Pump Genes	Genes	Positive Result
AMC-resistant isolates	*MexA*	178 (89%)
	*MexB*	178 (89%)
	*OprM*	178 (89%)
	*MexR*	178 (89%)
	*MexC*	178 (89%)
	*MexD*	178 (89%)
	*OprJ*	178 (89%)
	*NfxB*	178 (89%)

**Table 7 antibiotics-12-00486-t007:** Non-synonymous mutation of the *mexA* gene.

Codon Position	Reference Amino Acid	Altered Amino Acid	Amino Acid Position
389	GGT (Glycine)	AGT (Serine)	368

**Table 8 antibiotics-12-00486-t008:** *mexA* Prediction result of I-Mutant software.

Wild Type	New	I-Mutant Prediction Effect	DDG Value	Reliability Index (RI)	Temperature	pH
G (Glycine)	S (Serine)	Decrease	−1	8	25	7

**Table 9 antibiotics-12-00486-t009:** Synonymous and non-synonymous mutations of the mexB gene.

Codon Position	Reference Amino Acid Position	Altered Amino Acid Position	Amino Acid Position
Synonymous mutation of *mexB* gene			
148	TCC-TCG	Serine	129
154	AGC-AGT	Serine	130
184	GTC-GTG	Valine	142
256	CCT-CCG	Proline	166
259	CTC-CTA	Leucine	167
302	AAA-AAG	Lysine	290
308	GTA-GTC	Valine	291
635	CAA-CAG	Glutamine	673
Non-synonymous mutation of the *mexB* gene			
126	Asparagine (AAC)	Aspartate (GAC)	123
129	Tyrosine (TAT)	Asparagine (AAT)	124
136	Leucine (CTC)	Arginine (CGC)	126
138	Phenylalanine (TTC)	Tyrosine (TAC)	127
140	Phenylalanine (TTC)	Isoleucine (ATC)	128
151	Aspartate (GAC)	Glutamate (GAG)	131
165	Alanine (GCC)	Glycine (GGC)	136
167	Cysteine (TGC)	Serine (AGC)	137
170	Proline (CCG)	Methionine (ATG)	138
191	Glutamine (CAA)	Glutamate (GAA)	145
197	Leucine (CTC)	Glycine (GGC)	147
200	Proline (CCC)	Threonine (ACC)	148
203	Asparagine (AAC)	Aspartate (GAC)	149
215	Proline (CCC)	Alanine (GCC)	143
219	Leucine (CTG)	Glutamine (CAG)	154
228	Alanine (GCC)	Valine (GTG)	157
231	Leucine (CTC)	Glutamine (CAG)	158
244	Histidine (CAC)	Glutamine (CAA)	162
269	Glutamine (CAA)	Glutamate (GAA)	171
283	Histidine (CAT)	Glutamine (CAG)	175
292	Histidine (CAC)	Arginine (CGG)	287
303	Serine (TCG)	Alanine (GCG)	291
321	Leucine (CTG)	Methionine (ATG)	296
324	Leucine (CTG)	Valine (GTG)	298
327	Leucine (CTG)	Valine (GTG)	299
330	Arginine (CGT)	Glycine (GGT)	300
340	Proline (CCT)	Valine (GTT)	302
365	Asparagine (AAC)	Lysine (AAG)	311
378	Histidine (CAC)	Asparagine (AAC)	316
388	Alanine (GCT)	Valine (GTT)	319
424	Alanine (GCC)	Glycine (GGC)	331
429	Cysteine (TGC)	Glycine (GGT)	333
439	Proline (CCG)	Glutamine (CAG)	336
441	Leucine (CTG)	Valine (GTG)	337
456	Histidine (CAC)	Tyrosine (TAC)	342
488	Asparagine (AAT)	Lysine (AAG)	472
536	Histidine (CAT)	Glutamine (CAG)	488
590	Asparagine (AAC)	Lysine (AAG)	506
599	Histidine (CAT)	Tyrosine (CAG)	509
732	Histidine (CAT)	Tyrosine (CAG)	673

**Table 10 antibiotics-12-00486-t010:** *MexB* gene Prediction results of I-Mutant software.

Wild Type	New Type	I-Mutant Prediction Effect	DDG Value	Reliability Index (RI)	Temperature	pH
N	D	Decrease	−0.95	7	25	7
Y	N	Increase	−0.24	0	25	7
L	R	Decrease	−0.95	7	25	7
F	Y	Decrease	−0.85	7	25	7
F	I	Decrease	−1.99	9	25	7
D	E	Decrease	−0.59	7	25	7
A	G	Decrease	−1.03	7	25	7
C	S	Decrease	−0.53	1	25	7
P	M	Decrease	−0.96	1	25	7
Q	E	Decrease	−0.29	4	25	7
L	G	Increase	0.22	2	25	7
P	T	Decrease	−0.02	1	25	7
N	D	Increase	0.11	5	25	7
P	A	Decrease	−1.02	4	25	7
L	Q	Decrease	0.14	1	25	7
A	V	Decrease	−0.93	6	25	7
L	Q	Decrease	0.00	3	25	7
H	Q	Decrease	−0.61	7	25	7
Q	E	Decrease	−0.11	1	25	7
H	Q	Decrease	−0.61	7	25	7
H	R	Decrease	−1.37	9	25	7
S	A	Decrease	−0.90	8	25	7
L	M	Decrease	−0.80	5	25	7
L	V	Decrease	−1.30	6	25	7
L	V	Decrease	−1.32	6	25	7
R	G	Decrease	−0.48	1	25	7
P	V	Decrease	−1.57	4	25	7
N	K	Increase	−0.48	3	25	7
H	N	Decrease	−0.66	9	25	7
A	V	Decrease	−1.37	7	25	7
A	G	Increase	−0.51	1	25	7
C	G	Decrease	−0.76	0	25	7
P	Q	Decrease	−0.41	6	25	7
L	V	Decrease	−0.74	4	25	7
H	Y	Decrease	0.04	1	25	7
N	K	Increase	0.04	4	25	7
H	Q	Decrease	−0.53	6	25	7
N	K	Decrease	−0.55	2	25	7
H	Q	Decrease	−0.97	8	25	7
H	Q	Decrease	−0.91	6	25	7

**Table 11 antibiotics-12-00486-t011:** Synonymous and non-synonymous mutations of the *oprM* gene.

Codon Position	Reference Amino Acid Position	Altered Amino Acid Position	Amino Acid Position
Non-synonymous mutation of the OprM gene			
**11**	Glutamine (CAA)	Arginine (CGC)	7
**50**	Valine (GTG)	Alanine (GCG)	20
Synonymous mutation of the *OprM* gene			
**43**	ACT-ACC	T	17

**Table 12 antibiotics-12-00486-t012:** *OprM* gene Prediction results of I-Mutant software.

Wild Type	New Type	I-Mutant Prediction Effect	DDG Value	Reliability Index (RI)	Temperature	PH
Q (Glutamine)	R (Arginine)	Increase	−0.11	1	25	7
V (Valine)	A (Alanine)	Decrease	−1.66	8	25	7

**Table 13 antibiotics-12-00486-t013:** Docking energy score in kcal/mol.

Complex	Docking Score
max-A wild_meropenem	−6.1
max-A mutant (E178K) meropenem	−6.5
max-B wild_meropenem	−5.7
max-B mutant_meropenem	−8

## Data Availability

The data generated in the research study is presented in the manuscript.

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
