# Peer review of "Identification of Efflux Pump Mutations in Pseudomonas aeruginosa from Clinical Samples"

_antibiotics, 2023, doi:10.3390/antibiotics12030486_

Round 1

Reviewer 1 Report

Abstract too long. Delete list of mutations in mexB and replace with many mutations. The ligands used for docking with mexA and mexB are not mentioned. Moreover, there is no indication of the mexB mutant(s) used for docking among all those identified.

Results.

Figure presents the photo of the gel of PCR products in relation to the identification of the strains. Figure 2 to 9 of the PCR gels bring nothing and presentation of only 9 isolats on 200 without identification. These PCRs show the presence of mex genes in the clinical isolates, which must be the case for all P. aeruginosa. On the other hand, it would have been wiser to check by RT-PCR whether these genes are expressed and at what rate in isolates presenting multiresistance to antibiotics, in particular to CTX, CFM, TCG, SXT.

Mutation analysis. Table 6 : There is an inconsistency between the position of the nucleotides and the position of the amino acid. I think it should read "codon position" instead. 

Table 9: there is a contradiction between what is announced in materials and methods on the results of I-mutant analysis "DDG positive increase, DDG negative decrease' and the table where negative values are annoted increase and positive values decrease.

The tables are presented in bulk without any real analysis or summary of the results. To simplify the whole, it is better to put these tables in the appendix. A title in full of the acronyms used would have facilitated reading.

The "Dynamut and docking score" tables are incorrectly numbered. Docking; which mexB mutant was used? The docking figures are difficult to interpret because the ribbon representation of the wild type and mutant proteins shows different orientations. In the representation of wild type mexB the transmembrane domains do not appear. We even have the impression of dealing with 2 different proteins.

Conclusion: It is not possible to say that the clinical strains of P. aeruginosa strongly resistant to antibiotics "express all the genes encoding the efflux pumps" because this was not verified in the study. Detection of the presence of the genes of the mexANoprM and mexCDoprJ systems is presented but not that of measurement of their expression rate.

This article cannot be published as is. It needs to be heavily reworked to give consistency to the whole and to bring out the points concerning mutations, in particular in mexB, with their effects on the increase or decrease in the binding o antibiotics and their efflux by treating only the isolates the most resistant. The introduction should be more documented and clearly define the purpose of the study. The docking part must be linked to the mutation part in line with the numbering of figures and tables. the writing must be revised because the text has inadequate punctuation.

Author Response

REPLY TO THE COMMENTS

The authors are very thankful to the reviewers for their valuable comments on our manuscript and we are sure that it will polish our manuscript.  Following are para wise reply to the comments of worthy reviewers and the changes has been highlighted in the revised manuscript. 

REVIEWER 1

Query 1: Abstract too long. Delete the list of mutations in mexB and replace it with many mutations. The ligands used for docking with mexA and mexB are not mentioned. Moreover, there is no indication of the mexB mutant(s) used for docking among all those identified.

Answer: The abstract has been revised in light of the corrections suggested by the worthy reviewer.  Added to the revised manuscript abstract section.

Query 2: Figure presents the photo of the gel of PCR products in relation to the identification of the strains. Figures 2 to 9 of the PCR gels bring nothing and presentation of only 9 isolates on 200 without identification. These PCRs show the presence of mex genes in the clinical isolates, which must be the case for all P. aeruginosa. : On the other hand, it would have been wiser to check by RT-PCR whether these genes are expressed and at what rate in isolates presenting multi-resistance to antibiotics, in particular to CTX, CFM, TCG, and SXT.

Answer: In the current research study, a total of 200 resistant isolates of P. aeruginosa were identified both phenotypically and genetically. These PCR gels of mex genes are representative images for all isolates. The aim of this study was to investigate the role of the mexAB-oprM and mexCD-oprJ efflux pump system in antibiotic resistance in isolates of P. aeruginosa. The unavailability of the Real-Time PCR lead us to use conventional PCR and it is economical to use conventional PCR in mutational studies.    

Query 3: Mutation analysis. Table 6 : There is an inconsistency between the position of the nucleotides and the position of the amino acid. I think it should read "codon position" instead.

Answer: The suggested changes have been made in the revised manuscript.

Query 4: Table 9: there is a contradiction between what is announced in materials and methods on the results of I-mutant analysis "DDG positive increase, DDG negative decrease' and the table where negative values are annotated increase and positive values decrease.

Answer: In the materials and methods section the sentence has been rephrased. Moreover, DDG indicates the value of energy either in form of negative or positive. The term Increase is used for stability of mutation while the term Decrease indicates harmful or disease-causing effect of the mutation.

Query 5: The tables are presented in bulk without any real analysis or summary of the results. To simplify the whole, it is better to put these tables in the appendix. A title in full of the acronyms used would have facilitated reading.

Answer: A table has been shifted to supplementary materials as desired by the reviewer 

Query 6: The "Dynamut and docking score" tables are incorrectly numbered. Docking; which mexB mutant was used? The docking figures are difficult to interpret because the ribbon representation of the wild type and mutant proteins shows different orientations. In the representation of wild type mexB the transmembrane domains do not appear. We even have the impression of dealing with 2 different proteins.

Answer:  It is revisited in the revised manuscript and corrections are done.

Query 7: Conclusion: It is not possible to say that the clinical strains of P. aeruginosa strongly resistant to antibiotics "express all the genes encoding the efflux pumps" because this was not verified in the study. Detection of the presence of the genes of the mexANoprM and mexCDoprJ systems is presented but not that of measurement of their expression rate.

Answer: The sentence has been rephrased to give a clear picture of our findings. "express all the genes encoding the efflux pumps" means the expression of the selected efflux pump genes 

Query 8: This article cannot be published as is. It needs to be heavily reworked to give consistency to the whole and to bring out the points concerning mutations, in particular in mexB, with their effects on the increase or decrease in the binding of antibiotics and their efflux by treating only the isolates the most resistant. The introduction should be more documented and clearly define the purpose of the study. The docking part must be linked to the mutation part in line with the numbering of figures and tables. the writing must be revised because the text has inadequate punctuation.

Answer: The introduction has been supplemented to provide a clear purpose of the study.

Reviewer 2 Report

Thank you for giving me a chance to review this manuscript. Its very interesting topic. I have some comments to be considered for getting this manuscript in shape to be published.

- The research question is not clear to the reader. I suggest to add "Why its important to investigate the Efflux Pump and its role in P. aeruginosa resistance mechanism?" This can be at the end of the introduction part.

- Materials and Methods must be placed after the introduction. This will give the reader a glace of what type of results are expected.

- Results should be aggregated in logic order. i.e type of samples them demographic then microbiological, molecular and so on

- Some acronymous need to be fully described when written for the first time like KTH and HMC in line 297 !

- Be consistent in result part when presenting table . Like parameter , frequency and percentage (i.e Table 1 vs Table 5). unify the table layout.

All the best

Author Response

REPLY TO THE COMMENTS

The authors are very thankful to the reviewers for their valuable comments on our manuscript and we are sure that it will polish our manuscript.  Following are para wise reply to the comments of worthy reviewers and the changes has been highlighted in the revised manuscript. 

REVIEWER 2

Query 1: The research question is not clear to the reader. I suggest to add "Why its important to investigate the Efflux Pump and its role in P. aeruginosa resistance mechanism?" This can be at the end of the introduction part.

Answer: The suggested correction has been incorporated into the introduction part.

Query 2: Materials and Methods must be placed after the introduction. This will give the reader a glace of what type of results are expected.

Answer: The suggested changes have been made in the revised manuscript.

Query 3: Results should be aggregated in logic order. i.e type of samples them demographic then microbiological, molecular and so on

Answer: All the tables have been re-arranged according to the suggested order.

Query 4: Some acronymous need to be fully described when written for the first time like KTH and HMC in line 297

Answer: The suggested changes have been made in the revised manuscript.

Query 5: Be consistent in result part when presenting table. Like parameter, frequency and percentage (i.e Table 1 vs Table 5). Unify the table layout.

Answer: The Layout of the table has been re-arranged as suggested.

Round 2

Reviewer 1 Report

No comment

Author Response

Dear Reviewer, 

Thank you for the useful suggestions and accepting the manuscript for publication.